# Heart rate variability analysis for the prediction of pre-arousal during propofol-remifentanil general anaesthesia: A feasibility study

**Anne Wojtanowski**[1,2]*, **Maxence Hureau**[3,4], **Camille Ternynck**[5], **Benoit Tavernier**[2,3], **Mathieu Jeanne**[1,3,4], **Julien de Jonckheere**[1,2]

**1** CIC IT 1403, CHU Lille, Lille, France, **2** ULR 2694—METRICS, Univ. Lille, Lille, France, **3** Anesthésie Réanimation CHU Lille, CHU Lille, Lille, France, **4** ULR 7365 - GRITA - Groupe de Recherche sur les Formes Injectables et les Technologies Associées, Univ. Lille, Lille, France, **5** Département de Biostatistique, CHU Lille, Lille, France

* anne.wojtanowski@chu-lille.fr

**Data Availability Statement:** All relevant data are within the manuscript and its Supporting Information files.

## Abstract

Accidental awareness during general anaesthesia is a major complication. Despite the routine use of continuous electroencephalographic monitoring, accidental awareness during general anaesthesia remains relatively frequent and constitutes a significant additional cost. The prediction of patients' arousal during general anaesthesia could help preventing accidental awareness and some researchers have suggested that heart rate variability (HRV) analysis contains valuable information about the patient arousal during general anaesthesia. We conducted pilot study to investigate HRV ability to detect patient arousal. RR series and the Bispectral Index™ (BIS™) were recorded during general anaesthesia. The pre-arousal period T0 was defined as the time at which the BIS™ exceeded 60 at the end of surgery. HRV parameters were computed over several time periods before and after T0 and classified as "BIS™<60" or "BIS™≥60". A multivariate logistic regression model and a classification and regression tree algorithm were used to evaluate the HRV variables' ability to detect "BIS™≥60". All the models gave high specificity but poor sensitivity. Excluding T0 from the classification increased the sensitivity for all the models and gave AUCROC>0.7. In conclusion, we found that HRV analysis provided encouraging results to predict arousal at the end of general anaesthesia.

## Introduction

Accidental awareness during general anaesthesia (AAGA) is a major complication of anaesthesia and can cause long-term psychiatric problems. The incidence of AAGA ranges from 1 per 15,000 general anaesthesia (GA) procedures (when reported by anaesthesiologist) to 1–2 per 1000 (when reported by patients) [1–3]. The risk factors for AAGA include the type of surgery, demographic characteristics, the specific phase of anaesthesia (induction and emergence) and,

**Funding:** The author(s) received no specific funding for this work.

**Competing interests:** I have read the journal's policy and the authors of this manuscript have the following competing interests: Julien De Jonckheere and Mathieu Jeanne are shareholders and scientific consultants for MDoloris Medical Systems. The remaining authors have disclosed that they do not have any potential conflicts of interest.

most clearly, the use of neuromuscular blocking agents [4]. Several strategies for avoiding AAGA have been suggested. Clinical monitoring can be used to detect the occurrence of light anaesthesia, although neuromuscular blockade clearly interferes with this approach [5]. Another option is electroencephalographic (EEG) monitoring, in order to track the patient's level of consciousness via the Bispectral Index$^{TM}$ (BIS$^{TM}$, Medtronic, Minneapolis, MN, USA), the SedLine® (Masimo, Irvine, CA, USA) or entropy monitoring (GE Healthcare, Chicago, IL, USA). EEG monitors are considered as the gold standard to measure the level of consciousness and to detect AAGA. However, the increasing use of neuromuscular blocking agents in modern anaesthesia significantly impacts these measures. A study of myorelaxed -only volunteers showed that the use of neuromuscular blocking agents led to a decrease in EEG indexes like the BIS$^{TM}$ [6]. Similar results have been published with other EEG monitors [7]. Simplified EEG signal used for depth of hypnosis monitoring during general anaesthesia rely on a limited number of frontal electrodes, which capture the electrical potential of cortical and subcortical layers [8]. Propofol mediated unconsciousness has been related to an alpha/delta pattern and slow oscillations in the EEG [9, 10], but some authors found evidence that connected consciousness sometimes occurs in spite of these patterns being present [11]. Sensory disconnection and loss of consciousness may have different EEG spectral markers, with anterior and posterior cingulate regions probably playing a major role in consciousness [12]. Although a BIS$^{TM}$ below 60 is typically targeted during GA, one cannot be absolutely sure that awareness (i.e. AAGA) is precluded in this context. Although several studies have shown that the use of the BIS$^{TM}$ significantly reduced the occurrence of AAGA [13], recent evidence shows that awareness remains possible even when the EEG indexes are on target during anaesthesia [14].

Blood pressure monitoring and electrocardiography (ECG) are mandatory during GA. Acquisition of the ECG signal enables the evaluation of heart rate variability (HRV), which is a valid, well-established, non-invasive index of autonomous nervous system (ANS) responses during GA. For example, the Analgesia Nociception Index (ANI) is an HRV-based index of the nociception/antinociception (NAN) balance during GA [15, 16]. This technology is now used routinely to personalize analgesia: ANI monitoring has been shown to reduce or dynamically modify the use of opioids during and after surgery [17–19].

Many ANS monitors are used in clinical practice to assess the NAN balance and some of these ANS monitor are based, as a part, on HRV analysis [20]. On the other hand, hypnotic drugs may also affect HRV notably through its sympatholytic effect [21, 22]. In their study, R Huhle et al. demonstrated that propofol induction significantly reduced HRV features [23]. Base on this statement, Zhan J. et al. developed a HRV-derived system to evaluate the level of consciousness during GA and demonstrated good agreement between HRV and depth of anaesthesia [24]. We hypothesized that HRV is correlated with EEG signals of cortical awakening (i.e. when the BIS$^{TM}$ exceeds 60 but the patient is not yet conscious) at the end of GA. Hence, the objectives of the present ancillary, retrospective analysis of data collected during a prospective, single-centre, randomized clinical trial were to measure various HRV variables during propofol-remifentanil anaesthesia and to test the ability of an HRV-based model to predict "BIS$^{TM}$≥60" periods.

## Materials and methods

### Ethics

All methods were carried out in accordance with relevant guidelines and regulations. The study data were generated during the randomized ANI REMI LOOP clinical study ("Automatic control of ANI-guided remifentanil administration during propofol general anaesthesia"). This study aimed to demonstrate the efficacy and safety of an original closes-loop

remifentanil administration medical device (ANI-loop) in comparison to the standard practice. This study demonstrated that the use of the ANI-loop device significantly reduced the total amount of remifentanil administered. The study was registered at clinicaltrials.gov (NCT03556696) and with the French National Agency for Medicines and Health Product Safety (IDRCB 2017-A00858-45) and was conducted at Lille University Hospital between 27th June 2018 and 27th June 2022 for which patients gave a written informed consent. The present study is a retrospective analysis of the ANI REMI LOOP data. Under French regulations, studies concerning data reuse do not require additional authorisation by an ethical committee and only require informing participants. All patients were contacted and none of them objected to the use of their data for the study. The data were accessed after the completion of the ANI REMI LOOP study on September 2022. Authors had access to information that could identify participants during data collection but not during this data analysis.

## Procedures

The participants in the ANI REMI LOOP trial had undergone skin graft surgery under GA because of an accidental burn. Only adult patients with a sinus rhythm and no history of dysautonomia or diabetes mellitus were included in the ANI REMI LOOP study. Patients who presented with a non-sinus rhythm, had a pacemaker, heart transplant, baseline haemodynamic measures considered as hypotension or bradycardia or the opposite, heart rate baseline $>120\text{min}^{-1}$ or systolic blood pressure baseline $>160$ mmHg, were excluded. All patients were monitored throughout the procedure with standard techniques, including ECG, plethysmography, non-invasive blood pressure monitoring, and spirometry. Furthermore, all patients were monitored with the PhysioDoloris® device (V1.4.1.0, MDoloris, Loos, France) and the BIS™ VISTA system (Medtronic, Minneapolis, USA). The standardized anaesthetic protocol comprised midazolam ($0.08$ mg.kg$^{-1}$), remifentanil and propofol for the induction and maintenance of GA using a target-controlled infusion device (Orchestra® Base, Fresenius Kabi AG, Bad Homburg, Germany). The remifentanil and propofol targets were initially $6.0$ ng.ml$^{-1}$ and $5.0$ µg.ml$^{-1}$, respectively. Propofol was adjusted during GA so that the BIS™ stayed in the range [40–60] and remifentanil was adjusted so that the ANI stayed in the range [50–70]. Once myorelaxation had been achieved (cisatracurium, $0.15$ mg.kg$^{-1}$), the trachea was intubated, and the propofol and remifentanil targets were lowered to $3$ µg.ml$^{-1}$ and $3$ ng.ml$^{-1}$, respectively. All patients were ventilated using assisted controlled ventilation with tidal volume set at $8$ ml.kg$^{-1}$ of ideal body weight, and respiratory rate was set of $12$ min$^{-1}$ (GE Aisys anesthesia machine). The fraction of end-tidal (Fet) CO2 was maintained in the $30–35$ mmHg range. Propofol effect-site target was adjusted manually using the Schiller model. The remifentanil effect-site target was adjusted manually using the Minto model in the ANI REMI LOOP trial's control group and automatically in the interventional group. During the procedure, neuromuscular blockade was monitored by a train of four (TOF) stimulation [25]. At the end of the intervention, propofol and remifentanil were stopped only if the TOF ratio was up to 90%. Pre-emptive analgesia consisted of the local administration of ropivacain $2$ mg.ml$^{-1}$ during the procedure and paracetamol $1$ g and sufentanil $0.12$ µg.kg$^{-1}$ (maximum: $10$ µg) at the end of the procedure. Patients from whom the BIS™ recordings did not include the awakening phase were excluded for this retrospective analysis. Recordings showing synchronization problems between HRV and BIS™ data were not analyzed.

## ECG signal processing

The study database included the BIS™ and PhysioDoloris® recordings. Post-hoc analysis of the RR data exported from the PhysioDoloris® monitor enabled us to compute the HRV

variables. Ectopic beats and RR series artefact were detected and replaced by a linear interpolation using a specific algorithm [26, 27]. After RR series artefact filtering, the various HRV variables computed were:

- Time analysis:

  ○ The root mean square of successive differences (RMSSD) between adjacent RR intervals was computed over 100 RR intervals (approximately 2 minutes.). The RMSSD corresponds to short-term variations in the parasympathetic nervous system's activity [28].

  ○ The standard deviation of normal to normal (SDNN) RR intervals was computed over 100 RR intervals, which correspond to approximately 2 minutes. The SDNN corresponds to the overall variability of the ANS's activity.

  ○ Short-term variability (STV) was computed over 1 minute. After 4-Hz resampling of the RR series, the 1-minute window was divided into sixteen 3.75 s epochs. The average RR value was computed for each epoch. STV was defined as the mean difference between successive 3.75 s periods. STV corresponds to short-term variations in the parasympathetic nervous system's activity.

  ○ Long-term variation (LTV) was computed over 1 min. After 4-Hz resampling of the RR series, the 1-minute window is divided into sixteen 3.75 s epochs (i.e. 16 epochs). The average RR value was computed for each epoch. LTV was defined as the difference between the maximum and minimum values of the 16 epochs. LTV corresponds to the overall variability in the ANS's activity.

  Although STV and LTV are mainly used for fetal heart rate variability analysis, they use different methods than SDNN and RMSSD to assess the overall and short term variability of the ANS and could therefore provide additional information in the present clinical context.

- Spectral analysis:

  ○ Low frequency (LF) was based on spectral analysis of the RR series: for an adult, the standard bandwidth was defined as [0.05–0.15 Hz]. LF was computed via a wavelet transform (e.g. Daubechies 4-wavelet) on a 64-second window resampled at 8 Hz using a linear interpolation. LF reflects both sympathetic and parasympathetic modulations and is mainly influenced by baroreflex activity.

  ○ High frequency (HF) was based on spectral analysis of the RR series: for an adult, the standard bandwidth was defined as [0.15–0.4 Hz]. HF was computed in the same way as LF. HF reflects parasympathetic modulations only and is mainly influenced by respiratory sinus arrhythmia.

- Graphical analysis:

  ○ The ANI is a graphical index of the magnitude of HF oscillations. The RR series were resampled at 8 Hz, centred, and normalized in 64-second windows. The signal was band-pass filtered a wavelet transform in order to retain HF content only. The magnitude of these oscillations was measured as the surface area between the upper and lower envelopes. The ANI gives the relative magnitude of the HF spectral content on a 0-to-100 scale. The ANI is therefore a measure of parasympathetic activity and is mainly influenced by respiratory sinus arrhythmia.

  Each HRV measure was computed over a 60-second moving window with a 1-second sliding period and then averaged over 3 minutes.

Using the BIS signal as the existing gold standard, we defined T0 as the time at which the $BIS^{TM}$ exceeded 60 at the end of the surgery, i.e. after the remifentanil and propofol targets had been set to zero in order to awake the patient. We collected HRV data 5, 10, 20 and 30 minutes before T0 and 2 and 5 minutes after T0 defined respectively as T-5, T-10, T-20, T-30, T0, T+2 and T+5. The periods between these time points were rated as "$BIS^{TM}<60$" or "$BIS^{TM}\geq60$".

## Statistical analysis

The normality of distribution was assessed using histograms and the Shapiro-Wilk test. Quantitative variables were expressed as the mean ± standard deviation when normally distributed and as median [interquartile range (IQR)] when not. Qualitative variables were expressed as the number (percentage).

In order to identify highly correlated variables, a principal component analysis (PCA) followed by a varimax rotation was applied on all the available variables. Pearson's correlation coefficients between quantitative variables were described. Based on these results and clinical/physiological expertise, a set of uncorrelated variables was also selected. We first assessed the associations between all uncorrelated candidates predictors and the cortical awakening using univariable logistic regressions and secondly, we used multivariable method to develop a predictive model of cortical awakening. For each continuous predictor, the log-linearity assumption was assessed using restricted cubic spline functions. The absence of collinearity between variables was checked by calculating the variance inflation factors. When the log-linearity was rejected, variables were log-transformed. To account for the number of candidate predictors and limit the risk of over-optimism, this model was built using bootstrap resampling (n = 500) with, in each sample, automated stepwise backward selection procedure (with a removal criteria of p>0.05) from among all the uncorrelated candidate predictors [29]. A variable was kept in the final model if it was selected in at least 70% of these 500 analyses. The final predictive model was obtained by logistic regression of the retained variables. We computed the odds ratio (OR) and its 95% confidence intervals (CIs). The model's predictive ability was assessed as the area under the receiver operating characteristic curve (AUROC) [95%CI]. In parallel, we also developed a decision tree by using a classification and regression tree (CART) algorithm (Rpart package, R software) [30]. The models' respective levels of performance were assessed by calculating the sensitivity (Se), specificity (Sp), positive predictive value (PPV) and negative predictive value (NPV). The analyses were conducted on the complete set of cases. All tests were two-tailed, and the threshold for statistical significance was set to p<0.05. The data were analysed using the SAS software (version 9.4, SAS Institute Inc., Cary, NC, USA), and R software (version 3.6.1) [30] and Sipina research software (version 3.13).

## Results

A total of 52 patients were included in the initial clinical trial. Eleven of the 52 datasets lacked reliable synchronisation between the $BIS^{TM}$ and HRV signals, and 7 $BIS^{TM}$ recordings did not include the awakening phase (the electrode was removed before the end of the procedure). Hence, 34 datasets were available for the analyses and generated 208 measurements: 70 with "$BIS^{TM}\geq60$" and 138 with "$BIS^{TM}<60$" (for details, see Table 1). Table 2 describes the characteristics of the study participants, and Table 3 describes the characteristics of the HRV indices.

**Table 1. Time points and measurements.**

| Time point | Number of available measurements | Number of measurement with a BIS$^{TM}$ ≥ 60 |
|:---:|:---:|:---:|
| T-30 | 34 | 1 |
| T-20 | 34 | 0 |
| T-10 | 34 | 0 |
| T-5 | 34 | 1 |
| T0 | 34 | 34 |
| T+2 | 23 | 21 |
| T+5 | 15 | 13 |

Time points definition: T-30: 30 minutes before BIS$^{TM}$ ≥ 60; T-20: 20 minutes before BIS$^{TM}$ ≥ 60; T-10: 10 minutes before BIS$^{TM}$ ≥ 60; T-5: 5 minutes before BIS$^{TM}$ ≥ 60; T0: the moment that the BISTM ≥ 60; T+2: 2 minutes after BIS$^{TM}$ ≥ 60; T+5: 5 minutes after BIS$^{TM}$ ≥ 60

**Table 2. Demographic and clinical characteristics of the study participants.**

| Characteristic | |
|:---|:---|
| Sex, male | 13/34 (38.23) |
| Age, y | 41.50 [34.25–50.00] |
| Weight, kg | 75.59 ± 13.49 |
| Height, cm | 171.6 ± 8.16 |
| ASA score | |
| ASA I | 14/34 (41.18) |
| ASA II | 19/34 (55.88) |
| ASA III | 1/34 (2.94) |
| Duration of surgery, min | 48.09 ± 14.89 |
| Duration of anaesthesia, min | 91.76 ± 15.80 |
| Time between T0 and extubation, min | 5.43 [3.277–8.932] |
| Anaesthetic management | |
| propofol, mg | 682.5 ± 239.90 |
| remifentanil, µg | 757.0 [508.5–1006.8] |
| atropine | 2/34 (5.88) |
| ephedrine | 7/34 (20.58) |
| noradrenalin | 3/34 (8.88) |
| Analgesia management | |
| sufentanil, µg | 7.956 ± 1.13 |
| ANI REMI LOOP group | 18/34 (52.94) |
| Gabapentin | 6/33 (18.18) |
| Tramadol | 12/32 (37.5) |
| Paracetamol-opium | 4/31 (12.90) |
| Short-acting oral morphine | 3/30 (10) |
| Long-acting oral morphine | 3/30 (10) |
| Amitriptyline | 1/30 (3.33) |
| Ketoprofen | 0/32 (0) |

Values are expressed as the number/total number (%), the mean ± standard deviation, or the median [IQR].

ASA: American Society of Anaesthesiologists Physical Status Classification System

**Table 3. HRV variables, as a function of BIS$^{TM}$<60 and BIS$^{TM}$≥60 periods.**

| Variable | BIS$^{TM}$<60 (n = 138) | BIS$^{TM}$≥60 (n = 70) |
|---|---|---|
| BIS | 41.6 [34.4–47.2] | 64.7 [61.7–74.2] |
| HR | 64.79 ± 13.1 | 65.82 ± 13.86 |
| ANI | 68.78 ± 17.12 | 61.65 ± 16.13 |
| STV | 6.81 [4.55–9.85] | 9.03 [5.81–13.50] |
| SDNN | 24.27 [16.07–32.99] | 30.89 [20.65–44.49] |
| LTV | 64.52 [44.98–89.47] | 90.69 [60.46–127.14] |
| RMSSD | 16.94 [8.69–26.74] | 21.35 [13.61–27.69] |
| LF | 0.17 [0.05–0.35] | 0.29 [0.13–0.59] |
| HF | 0.03 [0.01–0.07] | 0.06 [0.02–0.13] |

Values are expressed as the number/total number (%), the mean ± standard deviation, or the median [IQR].
BIS: Bispectral Index$^{TM}$, HR:heart rate, ANI: Analgesia Nociception Index, STV: short-term variation, SDNN: standard deviation of RR intervals, LTV: long term variation, RMSSD: root mean square of successive difference in RR intervals, LF: low frequency, HF: high frequency

## Identification of uncorrelated variables

The PCA with varimax rotation demonstrated that SDNN, RMSSD, LTV and STV were highly correlated with the first factor, HF and LF were highly correlated with the second factor, and heart rate and ANI were highly correlated with the third factor (Fig 1 and Table 4).

Three sets of variables were strongly correlated with the three factors. Both the ANI and the heart rate were correlated with factor 3. Pearson's correlations coefficients between the variables were computed (Fig 1) and given that the ANI and the heart rate were weakly correlated with each other (Pearson's coefficient = -0.28), both variables were retained. For variables correlated with factor 1 (the SDNN, RMSSD, LTV and STV), we retained the SDNN and RMSSD. LTV and STV were highly correlated with the SDNN (Pearson's coefficient = 0.93 and 0.9, respectively). We retained the SDNN (rather than LTV and STV) because it is a well-known index of overall ANS activity; in contrast, LTV and STV were initially developed for foetal diagnosis and are only used in this particular clinical context [31, 32]. Although the RMSSD was also correlated with the SDNN (Pearson's coefficient = 0.82), we decided to keep it out of the model because it represents another autonomic pathway (the parasympathetic nervous system). Both LF and HF were correlated with factor 2 but we retained LF only; HF is solely influenced by parasympathetic activity, which is also accounted for by the ANI and the RMSSD. Hence, five variables were retained (given in bold in Table 4).

## Construction of cortical awakening prediction models

Shape of associations between each parameter and awakening were shown in Fig 2. There is no major deviation in log-linearity assumptions for all parameters, except for SDNN and LF where log-transformations were needed.

In univariate logistic regressions, the heart rate was the only variable that did not predict cortical awakening (Fig 3). The SDNN, RMSSD, LF and ANI were significant for the prediction of BIS$^{TM}$≥60, with AUROC of 0.65, 0.57, 0.64 and 0.64, respectively.

A multivariate logistic regression model with bootstrap resampling and stepwise backward selection (five variables) led to selection of the SDNN and the ANI. The AUROC of the model with these two variables was 0.74, which corresponds to a good level of discriminant power. By considering a threshold of 0.39, we obtained a sensitivity of 0.53, a specificity of 0.85, a PPV of 0.64, and an NPV of 0.78. However, when considering the 33 (out of 70) periods misclassified as

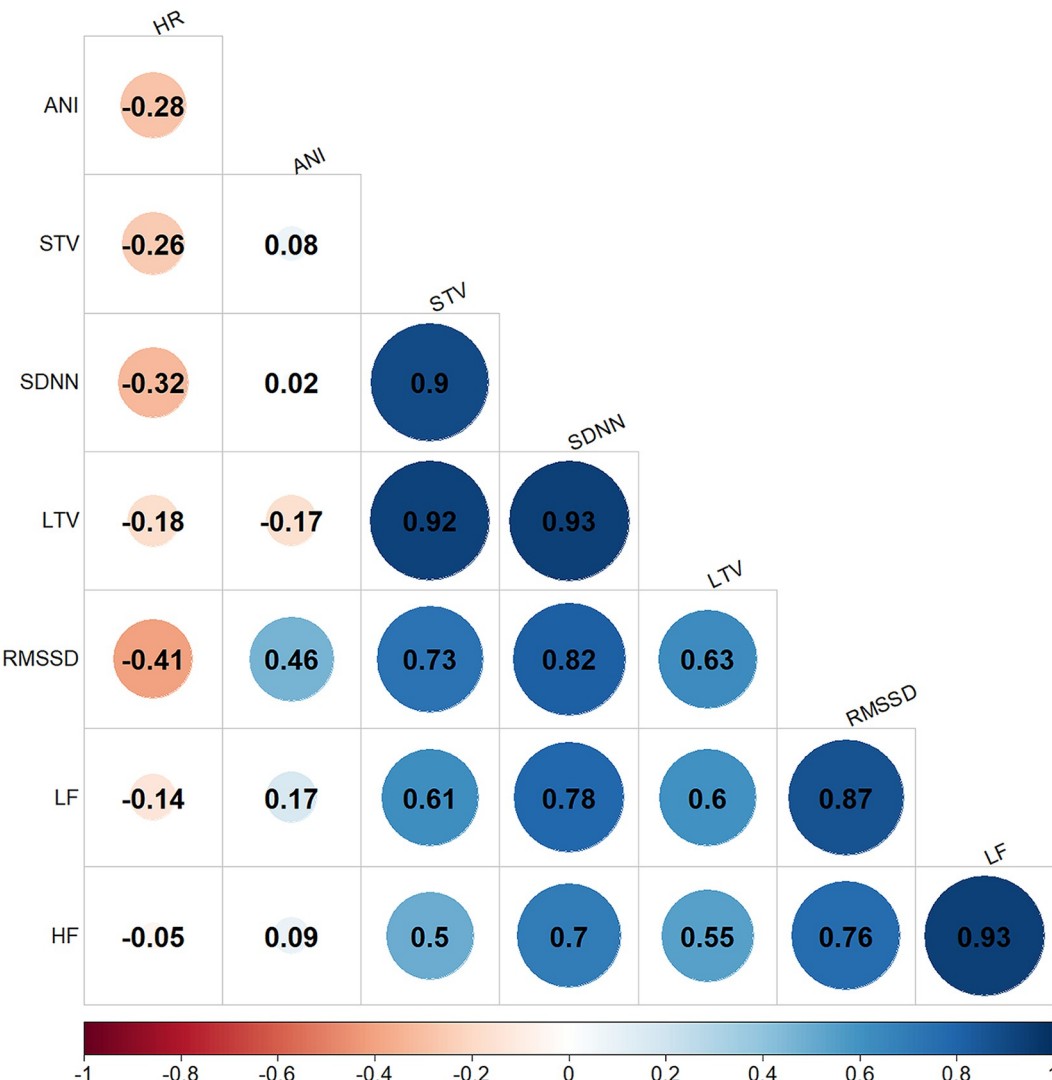

**Fig 1. Pearson coefficients for the correlations between the eight variables.** Variables were considered to highly correlated if Pearson's coefficient was > 0.70.

"BIS$^{TM}$≥60", 18 were related to T0 (Table 5). Excluding T0 from the classification increased the sensitivity to 0.58. For T2, 7 of the 21 "BIS$^{TM}$≥60" were misclassified, giving a specificity of 0.67.

Next, a CART decision tree algorithm was applied (Fig 4). The five uncorrelated variables were used in the algorithm. Using this algorithm, the variable that first classified the recordings as "BIS$^{TM}$≥60" or "BIS$^{TM}$<60" was the SDNN, followed by the ANI, LF, and the SDNN again (Fig 4). This model correctly classified 38 of the 70 "BIS$^{TM}$≥60" periods and 124 of the 138 "BIS$^{TM}$<60" periods, giving a sensitivity of 0.54, a specificity of 0.90, a PPV of 0.73, and an NPV of 0.79. However, when considering the 32 (out of 70) "BIS$^{TM}$≥60" periods misclassified as "BIS$^{TM}$<60", 20 were related to T0 (Table 6). Excluding T0 from the classification led to a sensitivity of 0.68. For T2, 16 of the 21 "BIS$^{TM}$≥60" periods were correctly classified, giving a sensitivity of 0.76.

**Table 4. Rotated factor loadings.**

|  | Factor 1 | Factor 2 | Factor 3 |
|---|---|---|---|
| **SDNN** | **0.94** | 0.26 | 0.10 |
| **RMSSD** | **0.81** | 0.17 | 0.28 |
| LTV | **0.61** | 0.03 | -0.26 |
| STV | **0.93** | 0.20 | 0.00 |
| **LF** | 0.21 | **0.97** | 0.03 |
| HF | 0.19 | **0.97** | -0.05 |
| **HR** | -0.15 | 0.01 | **-0.69** |
| **ANI** | -0.12 | 0.00 | **0.84** |

Absolute values greater than 0.6 are given in bold. Variables retained in the multivariable models are also given in bold. SDNN: standard deviation of RR intervals, RMSSD: root mean square of successive differences in RR intervals, LF: low frequency HR: heart rate, ANI: Analgesia Nociception Index.

## Discussion

This exploratory ancillary study of a number of HRV variables during awakening from propofol/remifentanil- GA was based on data recorded for 34 patients in the periods before and after the $\text{BIS}^{\text{TM}}$ crossed the threshold of 60; this usually occurs several minutes before the patient awakes. Our study's objective was to test the ability of an HRV-based model to predict "$\text{BIS}^{\text{TM}} \geq 60$" periods. We developed two multivariate models: the first using multivariate logistic regression, and the second using a CART algorithm.

### HRV analysis results interpretation

The regression analysis enabled us to evaluate each HRV variable's ability to predict "$\text{BIS}^{\text{TM}} \geq 60$" periods (Fig 3). Unsurprisingly, the heart rate did not predict the early stages of awakening (OR [95%CI]: 1.01 [0.98–1.03]); this was probably because many factors influence the sinus node during this phase of GA. In contrast, the SDNN, RMSSD, LF and ANI highlighted significant differences between "$\text{BIS}^{\text{TM}} < 60$" and "$\text{BIS}^{\text{TM}} \geq 60$" periods but were poor predictors (AUROC of 0.65, 0.57, 0.64 and 0.64 for SDNN, RMSSD, LF, and ANI, respectively). The poor performances of ANI and RMSSD are in line with our current understanding of parasympathetic monitoring, which is considered, in the particular context of surgery under GA, to mostly reflect the NAN balance. The results are also in agreement with Boselli et al.'s finding that the ANI was a good predictor of the hemodynamic response to surgical stimulation but a poor predictor of the level of sedation [16]. In their study, Aragón-Benedí, C et al. demonstrated an association between ANI and the Richmond Agitation-Sedation Scale (RASS) in sedated intensive care unit patients [33]. However, although significant, the correlation was weak (r = 0.246, p = 0.009).

The results of the PCA and Pearson's correlation test showed that less specific HRV measures (such as STV and LTV) were strongly correlated with the SDNN (Pearson coefficients of 0.90 and 0.93, respectively (Fig 1)); hence we selected the SDNN alone for the construction of a predictive model. From a clinical point of view, the SDNN is a well-known HRV variable but is usually computed over several hours; in contrast, we used very short time windows (approx. 2 minutes for computation and 3 minutes for averaging). The SDNN during GA has been linked to the general decrease in ANS variability under the influence of (in most cases) hypnotic drugs [34]. It is therefore not surprising that the SDNN increases once propofol administration has been discontinued, leading to cortical awakening in the absence of other hypnotic drugs (as was the case in our series). From a physiopathological standpoint, the SDNN

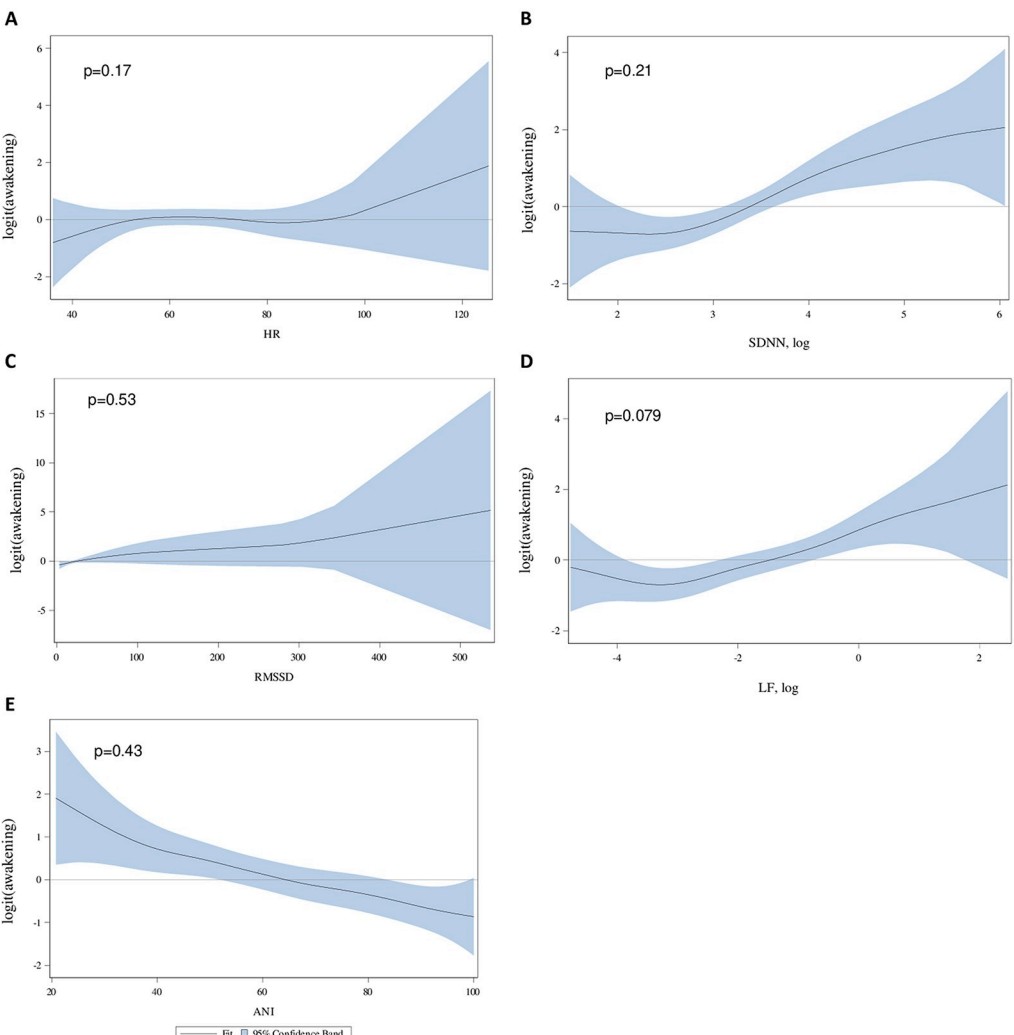

**Fig 2. Relationship between awakening and HRV variables.** P-values of the likelihood ratio test comparing the full model (including both nonparametric component and linear terms to the model including a linear term only.

measured over 2 minutes is likely to be influenced by both sympathetic and parasympathetic activities. Hence, the SDNN might represent the overall modulation of the ANS, whereas LF and HF reflect the HRV's information content in detail.

The RMSSD was strongly correlated with the SDNN (Fig 1), even though the latter gave high ORs during the awakening period; this observation shows how complex HRV measurements are. Interestingly, the LF spectral content was correlated strongly with the RMSSD (Pearson's correlation coefficient: 0.87 (Fig 1)). Like the SDNN, LF had a significant OR [95% CI] of 1.47 [1.17–1.84] (Fig 3).

Although the univariate analysis did not accurately predict BIS™≥60 periods, the multivariate logistic regression model had a AUROC of 0.74 and good specificity (0.85) but poor sensitivity (0.53). Decision trees generated with CART algorithm also gave high specificity (0.90) but gave poor sensitivities (0.54). Interestingly, the SDNN and the ANI were the most discriminant indexes in the two models. In the early stages of arousal, this discriminant ability might be related to activation of subcortical structures previously inhibited by hypnotic drugs.

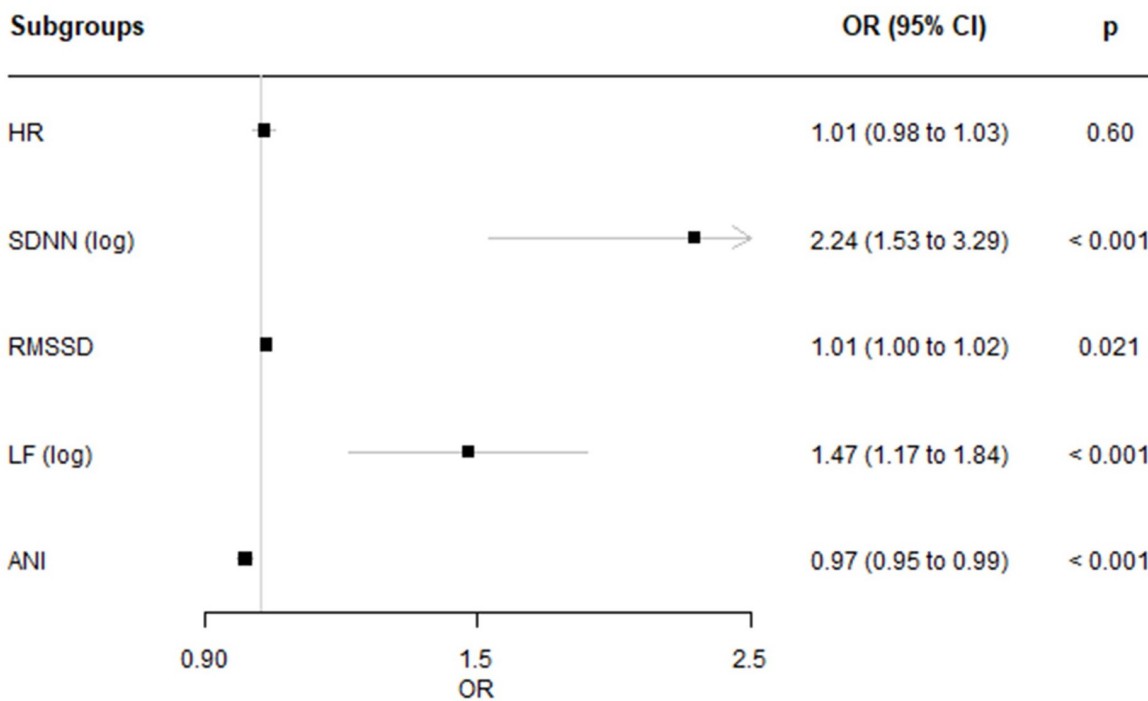

**Fig 3. Results of the univariate analysis for the prediction of BIS™≥60.**

By combining statistically significant, independent measurements and applying the CART algorithm, we produced a decision tree that described the study variables' respective abilities to predict awakening status (as measured by a BIS™ below or above 60). SDNN values appear on both the first and fourth branches of the decision tree. The ANI appears in the tree's second line. An important marker of the decision tree's performance is shown in Table 6: most of the misclassified values occurred at T0, when the BIS™ just exceeded 60. One must bear in mind that HRV measurements are averaged over three minutes; this necessarily induces a delay and might explain why the variables did not "see" the arousal. This hypothesis is strengthened by the greater sensitivity observed at T2 (vs. T0). The CART model's performance has clinical value because the patient was extubated a median of 5.4 [3.3–8.9] minutes after T0; hence, HRV measurements could alert the anaesthesiologists to the patient's imminent arousal.

To the best of our knowledge, the present feasibility study is one of the first to have combined exploratory statistical methods, the drastic selection of HRV variables, and clinical knowledge about the ANS in the context of GA. The study's main strength was the use of a well-documented dataset from a recent, standardized, prospective clinical trial in which propofol-remifentanil GA was guided by continuous monitoring of the BIS™ and ANI. Heart rate, blood pressure, motor blockade, NAN balance, and ventilator settings were recorded continuously. Pre-emptive analgesia was provided by a single bolus of sufentanil at the end of surgery. Lastly, in this ancillary study, we only assessed the end of the anaesthetic procedure in

**Table 5. Values misclassified by the regression model including the ANI and the SDNN.**

| Time | -30 | -20 | -10 | -5 | 0 | 2 | 5 |
|---|---|---|---|---|---|---|---|
| BIS™≥60 classified as BIS™<60 | 1 | 0 | 0 | 0 | 18 | 7 | 7 |
| BIS™<60 classified as BIS™≥60 | 7 | 7 | 2 | 5 | 0 | 0 | 0 |

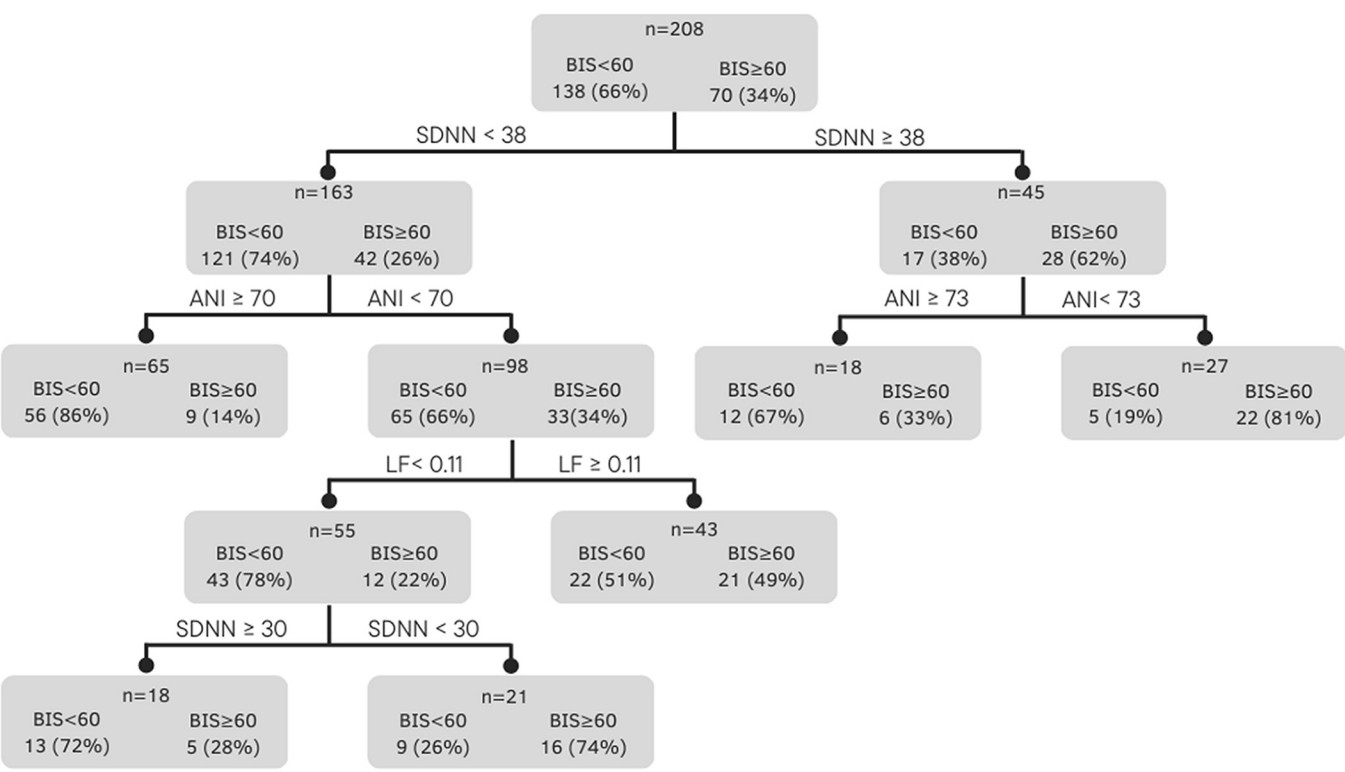

**Fig 4. The decision tree generated by the CART algorithm.** For each leaf, the predicted class is given in bold, the number of individuals in the BIS$^{TM}$≥60 class is given on the left, and the number of individuals in the BIS$^{TM}$<60 class is given on the right.

order to ensure the absence of neuromuscular blockage, which can negatively affect BIS$^{TM}$ measurements. Therefore, despite the known pitfalls of BIS monitoring, we are confident that HRV measurements were made at a time when connectedness was gradually restored in sub-cortical structures of the patients.

## Heart rate variability as a prognosis tool for unconsciousness

Researchers have used various strategies to try to determine whether HRV measurement can distinguish between various anaesthesia states (ranging from an awake state to unconsciousness): comparison of time-domain and frequency-domain variables in two anaesthesia states, construction of new index based on HRV, etc. The Similarity Index was described in 2007 as a time-domain marker of the similarity between consecutive data and was well able to distinguish between two states of isoflurane-induced anaesthesia (probability of correct prediction: 0.91) [35]. Similarly, two anaesthesia states differ significantly in their LF and HF spectra [35]. The non-rhythmic-to-rhythmic ratio (NRR) was defined in 2014 as the ratio between the momentary non-rhythmic power and the momentary rhythmic power; it provides insight into the magnitude of controlled, ventilatory-induced arrhythmia. The NRR performed better than

**Table 6. Values misclassified by the CART algorithm.**

| Time | -30 | -20 | -10 | -5 | 0 | 2 | 5 |
|---|---|---|---|---|---|---|---|
| BIS$^{TM}$≥60 classified as BIS$^{TM}$<60 | 1 | 0 | 0 | 0 | 20 | 5 | 6 |
| BIS$^{TM}$<60 classified as BIS$^{TM}$≥60 | 0 | 3 | 5 | 5 | 0 | 1 | 0 |

a gold standard in predicting a response to a nociceptive stimulus [36]. The Similarity and Distribution Index (SDI) described in 2017 is based on the same mathematical method as the Similarity Index and to which an artificial neural network model is additionally fitted. The SDI was strongly correlated with the state of deep anaesthesia, as defined by a panel of experienced anaesthesiologists. However, the application of the SDI is limited by the very heterogeneous anaesthetic protocols administered, which blurs the specific relationship with the depth of anaesthesia [37]. Other researchers have computed several HRV variables and fed them into a deep neural network; compared with an expert assessment, the overall accuracy for the depth of anaesthesia was 90.1%. However, the specific accuracy for awakening after GA was only 56.6%. Again, different anaesthesia protocols led to different conclusions [24].

The literature methods featured a variety of anaesthesia protocols (e.g. total intravenous anaesthesia, halogenated ethers, or even mixed protocols) and ran the models before the induction of anaesthesia and then through steady-state anaesthesia until awakening. We believe that inclusion of the induction period and the endotracheal tube placement during the modelled period is extremely challenging because the apnoea induced by induction has a strong influence on HRV variables in general and those related to parasympathetic activity (on which the influence of respiratory sinus arrhythmia is very strong) in particular [38].

## Limitations

The present study had some limitations. Firstly, the prospective ANI REMI LOOP clinical trial was well documented but was not designed for the specific objective of the post-hoc analysis presented here. Particularly, the sample size was small and not computed in accordance to the primary objective of this study. These results need to be confirmed on a larger population in a prospective clinical trial. Secondly, the amount of data was smaller during the arousal period (after T0) than during the time between T-30 and T0; this might have overloaded the models with "deep anaesthesia" data, relative to "pre-arousal" data. Lastly, we only analyzed simple, well-known time-domain and spectral HRV variables. The inclusion of more complex, non-linear indexes (e.g. the approximate entropy or the sample entropy) [39] would perhaps have led to better results. However, in order to facilitate physiopathological interpretation of our results, we decided to focus on variables known to be related to various aspects of the ANS. In this study we only considered HRV features whereas taking into account the blood pressure variability would allow obtaining a much more realistic model of the baroreflex response to anaesthetic agents [21, 22]. In future work, the addition of invasive or non-invasive continuous blood pressure monitoring could allow testing this approach.

## Conclusion

We found that HRV analysis provided accurate clinically useful insights into the "pre-arousal" recovery period of GA. Further prospective large scale clinical studies are necessary to validate these results and to confirm the ability of HRV measurements to predict AAGA.

## Supporting information

**S1 Table. Heart rate variability measures.** BIS: Bispectral Index[TM], HR:heart rate, ANI: Analgesia Nociception Index, STV: short-term variation, SDNN: standard deviation of RR intervals, LTV: long term variation, RMSSD: root mean square of successive difference in RR intervals, LF: low frequency, HF: high frequency.
(XLSX)

## Author Contributions

**Formal analysis:** Camille Ternynck.

**Investigation:** Anne Wojtanowski.

**Supervision:** Julien de Jonckheere.

**Validation:** Julien de Jonckheere.

**Writing – original draft:** Anne Wojtanowski, Maxence Hureau, Mathieu Jeanne, Julien de Jonckheere.

**Writing – review & editing:** Anne Wojtanowski, Maxence Hureau, Camille Ternynck, Benoit Tavernier, Mathieu Jeanne, Julien de Jonckheere.

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
