## [Decision Letter · Decision Letter 0]

21 Jun 2024

PONE-D-24-08727Heart rate variability analysis for the prediction of accidental awareness during general anaesthesia: a feasability studyPLOS ONE

Dear Dr. Wojtanowski,

Thank you for submitting your manuscript to PLOS ONE. After careful consideration, we feel that it has merit but does not fully meet PLOS ONE’s publication criteria as it currently stands. Therefore, we invite you to submit a revised version of the manuscript that addresses the points raised during the review process.

We look forward to receiving your revised manuscript.

Kind regards,

Ernesto Iadanza

Academic Editor

PLOS ONE

Journal Requirements:

Reviewers' comments:

Reviewer's Responses to Questions

**Comments to the Author**

1. Is the manuscript technically sound, and do the data support the conclusions?

Reviewer #1: Yes

Reviewer #2: Partly

Reviewer #3: Yes

Reviewer #4: Yes

Reviewer #5: Yes

2. Has the statistical analysis been performed appropriately and rigorously? 

Reviewer #1: Yes

Reviewer #2: N/A

Reviewer #3: Yes

Reviewer #4: Yes

Reviewer #5: Yes

3. Have the authors made all data underlying the findings in their manuscript fully available?

Reviewer #1: No

Reviewer #2: Yes

Reviewer #3: Yes

Reviewer #4: No

Reviewer #5: No

4. Is the manuscript presented in an intelligible fashion and written in standard English?

Reviewer #1: Yes

Reviewer #2: Yes

Reviewer #3: Yes

Reviewer #4: Yes

Reviewer #5: Yes

5. Review Comments to the Author

Reviewer #1: In the present work, two approaches for the prediction of accidental awareness during general anesthesia (AAGA) via the computation of HRV indexes are evaluated. Authors found that the proposed multivariate models have clinical value for anesthesiologists who can be alerted of imminent AAGA in their patients.

The manuscript is well-written and structured, highly readable with appropriate statistical analysis, good presentation of the results and discussion of the same that is supported by data. The clinical relevance of the problem is clearly stated and of interest for a clinical audience.

I have some minor comments:

1. In the Introduction an additional short paragraph could be added to better explain how HRV is affected by anesthesia. This would better bridge the background to the research hypothesis. I report as a suggestion some references (by DOI) that could be potentially a starting point for this discussion: 10.1093/bja/65.2.184; 10.1097/00000542-199205000-00010; 10.1152/japplphysiol.00537.2013; 10.3389/fphys.2019.01319

2. Table 1: please define T-30, -20, … etc. before the use in the Table and the Discussion chapter for clarity, and report it in the caption of the Table as well

3. Table 2: in my opinion the second section of the table (BIS<60 vs BIS>60) should be separated into a third table for readability

4. Lines 223-225: it is unclear to me LTV and STV were calculated at all if they were disregarded on the grounds of their classical clinical use. I agree with the authors on the selection on the indexes because I believe that a stronger adherence to the 1996 Task Force recommendations improves the strength of the manuscript.

5. Lines 357-358: add a reference to ApEn and SampEn.

6. In the limitations section, an interesting future development for the AAGA prediction could be the inclusion in the model of indexes derived from the blood pressure variability and from the bivariate analysis of the baroreflex, which are also affected by anesthesia (see previously mentioned references). Was invasive blood pressure monitoring performed during anesthesia in this database? If not, in future works it could be interesting to test this approach on a different database that includes this signal.

Reviewer #2: Title: Heart rate variability analysis for the prediction of accidental awareness during general anaesthesia: a feasability study

Journal: PLOS ONE

Manuscript ID: PONE-D-24-08727

General opinion

The authors measured various heart rate variability (HRV) variables during propofol-remifentanil anesthesia in clinical trials and tested the ability of an HRV-based model to predict “BISTM≥60” periods. They want to confirm that HRV analysis can be used to predict the occurrence of accidental awareness during general anesthesia. However, both experimental design and result analysis are far from achieving this hypothesis. The pictures are vague even the download version, some methods are not clear and no deeper analysis at all. Therefore, I do not recommend the publication of this paper. Comments are as follows:

Major comments

1. The title of this paper is overstated, they only used the ' propofol-remifentanil ' anesthesia program, which doesn’t represent all cases of general anesthesia.

2. The introduction section should include more recent studies on the emergence of accidental awareness during general anesthesia.

3. The narrative of the material and method part is not concise enough.

4. The writing was confused, and the methods and results do not correspond to each other.

5. The initial clinical trial included only 52 patients, and the sample size was too small to be convincing.

6. In the discussion section, they pointed out that they only analyzed simple, well-known time-domain and spectral HRV variables, the inclusion of more complex, non-linear indexes (e.g. the approximate entropy or the sample entropy) would perhaps have led to better results. It is recommended to add this part of the work in this study.

Minor comments

1. Please add the graded title before each part of the article.

2. The abstract part is not clear about the results and conclusions of this study.

3. Please unify the style of the table.

4. Please adjust the order of images and text, don’ t place the figure at the end of the article!

5. The resolution of the image is too low.

6. It would be better to include additional latest references.

7. Please revise the format of the references, the page number of the reference was listed (Page 23, line 385).

Reviewer #3: The research procedure is clear, and the result figure of the restricted cubic spline function can be added to the result part. The discussion part fails to explain well why HRV is used to evaluate the accidental awareness of general anesthesia.The figure at the end of the manuscript is not clear enough.

Reviewer #4: he article explores the feasibility of using heart rate variability (HRV) analysis to predict accidental awareness during general anaesthesia (GA). This pilot study addresses a significant complication in anaesthesia practice, where accidental awareness remains relatively frequent despite continuous electroencephalographic monitoring. By analyzing RR intervals and the Bispectral IndexTM (BISTM), the researchers aim to identify HRV parameters indicative of patient arousal.

The study demonstrates an innovative approach by leveraging HRV analysis, which is less commonly utilized in this context, to address the issue of accidental awareness. The methodology is comprehensive, employing both multivariate logistic regression and classification and regression tree algorithms to provide a robust analysis framework.

However, some issue that requires attention.

The phrase "Given that EEG signals are influenced by the brain’s cortex only and that awareness and connectedness result from interactions between subcortical structures, it is not surprising that EEG signals do not reflect the activation of the subcortex, which leads to awareness during surgery" requires clarification and proper referencing.

Firstly, the assertion that EEG signals are influenced by the brain’s cortex only is inaccurate. EEG signals, while predominantly reflective of cortical activity, can also capture electrical activity originating from subcortical structures through volume conduction. Furthermore, the claim that EEG signals do not reflect subcortical activation lacks a supporting reference.

ECG signal processing

I wonder whether and how the RR intervals were interpolated before applying spectral analysis? If interpolation was used, what kind of interpolation method was employed? Furthermore, how were potential ectopic beats treated during the analysis? Clarification on these methodological details is important for the reproducibility and reliability of the study's findings.

Statistical analysis

The phrase "Highly correlated variables were then identified using a principal component analysis (PCA) and then varimax rotation" is not entirely clear and may reflect a misunderstanding of PCA's purpose. To my knowlage, Principal component analysis (PCA) is not primarily a method for identifying highly correlated variables. Instead, PCA is a dimensionality reduction technique that transforms a set of possibly correlated variables into a smaller number of uncorrelated variables called principal components. These principal components are linear combinations of the original variables and aim to capture the maximum variance in the data.

While PCA can reveal underlying patterns and relationships in the data, it does not directly identify which variables are highly correlated. Instead, it identifies combinations of variables (principal components) that account for the most variance. Varimax rotation, a method used after PCA, is applied to the principal components to make the output more interpretable by maximizing the variance of the squared loadings of each factor, thereby simplifying the structure.

Reviewer #5: Dear Authors,

You have chosen a very interesting topic with high clinical relevance. The assessment of the pre-arousal and arousal with the energy of the autonomic nervous system and the heart rate variability with the monitor ANI is innovative. However, there are essential points of criticism that need addressing. The paper should be revised thoroughly. Please focus on the improvements suggested below, as I believe this work is worth it.

Best regards.

Major Changes:

1. Confusion between Accidental Awareness and Arousal: Throughout the article, there is confusion between "accidental awareness" and "arousal." The study aims to correlate ANI and other HRV variables with arousal or pre-arousal during anesthetic emergence, not specifically with accidental awareness during maintenance of anesthesia. This needs to be corrected in the title and throughout the article. For example, "Heart rate variability for the prediction of pre-arousal during general anaesthesia." Similarly, the first hypothesis and the statement "and could be useful to prevent accidental awareness during general anaesthesia" are not demonstrated with the current methodology and should be revised. The distinction between awareness and proprioceptive afference preceding consciousness needs to be clarified, especially in statements like "Therefore, despite the known pitfalls of BIS monitoring, we are confident that HRV measurements were made at a time when connectedness was gradually restored in subcortical structures of the patients." Agreeing with the conclusion, "we found that HRV analysis provided accurate clinically useful insights into the 'pre-arousal' recovery period of GA," is appropriate, but it does not predict AAGA or cortical awakening, only an increase in subcortical and cortical afference.

2. Methodology and Results of ANI REMI LOOP Study: The study is based on retrospective data from the ANI REMI LOOP clinical study, yet there is no published information about this trial. Provide more information on the methodology and results of this clinical trial.

3. Data Availability: The data availability section states "No - some restrictions will apply," but later mentions "The datasets used and/or analysed during the current study are available from the corresponding author on reasonable request." Please specify and ensure compliance with PLOS ONE guidelines.

4. EEG and EMG Interference: It is well-known that processed EEG like BIS is influenced by EMG, especially above 60, as stated, "Clinical monitoring can be used to detect the occurrence of light anaesthesia, although neuromuscular blockade clearly interferes with this approach." The study was conducted only at the end of surgery to "ensure the absence of neuromuscular blockage, which can negatively affect BIS measurements." If neuromuscular blockade (NMB) was not monitored, how can the increase above 60 be justified as not due to beta and gamma wave interference from EMG? Also, "Once myorelaxation had been achieved" does not specify if NMB was used, which drug and dose. Was NMB monitored? What type of NMB was used? Was the blockade reversed with neostigmine, sugammadex, or spontaneously? How was EMG interference adjusted?

5. Ventilation and ANI: Changes in ventilator pressures, NMB, and uncontrolled ventilation can decrease ANI values. How was this verified, particularly regarding "those related to parasympathetic activity (on which the influence of respiratory sinus arrhythmia is very strong) in particular [25]"? This needs to be addressed.

6. Adjustment Methods: It is unclear if BIS was adjusted by increasing or decreasing remifentanil and if ANI was adjusted with propofol. Specify this, "The remifentanil and propofol targets were initially 6.0 ng.ml-1 and 5.0 μg.ml-1, respectively, and were adjusted during GA so that the BISTM stayed in the range [40-60] and the ANI stayed in the range [50-70]." Additionally, the mode of TCI and whether it was effect-site or plasma-site should be specified.

7. Discussion on ANI and Sedation: The discussion is confusing. Initially, you state, "The poor performances of ANI and RMSSD are in line with our current understanding of parasympathetic monitoring, which is considered to mostly reflect the NAN balance. The results are also in agreement with Boselli et al.’s finding that the ANI was a good predictor of the hemodynamic response to surgical stimulation but a poor predictor of the level of sedation [10]." However, you later say, "To the best of our knowledge, the ANI is mostly related to the NAN balance but can also be influenced by the withdrawal of hypnotic drugs – drugs that are well known to potentiate the effects of opioids. ANI reflects the NAN balance (and not cortical awakening) and might be explained by the hypnotic drugs’ potentiation of opioids in the context of slow arousal from propofol anaesthesia." This is contradictory and needs clarification. In fact, other studies (Aragón-Benedí, C et al. 2022 https://doi.org/10.1038/s41598-022-25537-z) have demonstrated the relationship between sedation scales (RASS) and ANI. "In our study, the ANIm value correlated directly with the RASS scale, i.e., those patients with greater sedation according to the RASS scale had higher ANIm values."

8. Discussion Structure: The discussion is somewhat disorganized. Suggest structuring it into subsections.

Minor Changes:

1. Short Title: The short title should be more concise. For example, "HRV for Prediction of Arousal During GA."

2. Incidence of Accidental Awareness: Accidental awareness is not as frequent nowadays. Provide more updated data.

3. Clarification of ANI: The Analgesia Nociception Index (ANI) is not specific to the nociception/antinociception (NAN) balance during GA. It should be corrected to an ANS index.

4. Clarification of "some experts": The phrase "some experts" is vague. Specify with appropriate references.

5. Use of PhysioDoloris Device: Explain why the non-commercial PhysioDoloris device was used instead of the new version of ANI. This impacts replicability.

6. Exclusion of LTV and STV: Clarify why LTV and STV were collected initially if they were to be excluded later, "in contrast, LTV and STV were initially developed for foetal diagnosis and are only used in this particular clinical context [19, 20]."

Conclusion:

This manuscript has great potential and scientific relevance for understanding the ANS in anesthesiology. However, it requires major and minor revisions before acceptance. I am confident that with these revisions, the article will be a valuable contribution to the scientific community.

6. PLOS authors have the option to publish the peer review history of their article (what does this mean?). If published, this will include your full peer review and any attached files.

Reviewer #1: No

Reviewer #2: No

Reviewer #3: No

Reviewer #4: No

Reviewer #5: **Yes: **Cristian Aragón-Benedí

---

## [Author Response · Author response to Decision Letter 0]

2 Aug 2024

We thank the reviewers for their comments. All the comments were taken into account in the new version of the document. 

Reviewer #1: In the present work, two approaches for the prediction of accidental awareness during general anesthesia (AAGA) via the computation of HRV indexes are evaluated. Authors found that the proposed multivariate models have clinical value for anesthesiologists who can be alerted of imminent AAGA in their patients.

The manuscript is well-written and structured, highly readable with appropriate statistical analysis, good presentation of the results and discussion of the same that is supported by data. The clinical relevance of the problem is clearly stated and of interest for a clinical audience.

I have some minor comments:

1. In the Introduction an additional short paragraph could be added to better explain how HRV is affected by anesthesia. This would better bridge the background to the research hypothesis. I report as a suggestion some references (by DOI) that could be potentially a starting point for this discussion: 10.1093/bja/65.2.184; 10.1097/00000542-199205000-00010; 10.1152/japplphysiol.00537.2013; 10.3389/fphys.2019.01319

As requested we explained how HRV is affected by anesthesia. We have modified the paragraph introduction (lines 83-90), as follows: 

“Many ANS monitors are used in clinical practice to assess the NAN balance and some of these ANS monitor are based, as a part, on HRV analysis [20]. On the other hand, hypnotic drugs may also affect HRV notably through its sympatholytic effect [21,22]. In their study, R Huhle et al. demonstrated that propofol induction significantly reduced HRV features [23]. Base on this statement, Zhan J. et al. developed a HRV-derived system to evaluate the level of consciousness during GA and demonstrated good afreement between HRV and depth of anaesthesia [24]. We hypothesized that HRV is correlated with EEG signals of cortical awakening (i.e. when the BISTM exceeds 60 but the patient is not yet conscious) at the end of GA.”

2. Table 1: please define T-30, -20, … etc. before the use in the Table and the Discussion chapter for clarity, and report it in the caption of the Table as well

As requested we defined in the text (line 191) T-30, -20,… etc as follows: 

“We collected HRV data 5, 10, 20 and 30 minutes before T0 and 2 and 5 minutes after T0 defined respectively as T-5, T-10, T-20, T-30, T0, T+2 and T+5.”

We also added a caption to Table 1: “Time points definition: T-30: 30 minutes before BISTM ≥ 60; T-20: 20 minutes before BISTM ≥ 60; T-10: 10 minutes before BISTM ≥ 60; T-5: 5 minutes before BISTM ≥ 60; T0: the moment that the BISTM ≥ 60; T+2: 2 minutes after BISTM ≥ 60; T+5: 5 minutes after BISTM ≥ 60”.

3. Table 2: in my opinion the second section of the table (BIS<60 vs BIS>60) should be separated into a third table for readability.

As requested we have separated the Table 2 into 2 tables. 

4. Lines 223-225: it is unclear to me LTV and STV were calculated at all if they were disregarded on the grounds of their classical clinical use. I agree with the authors on the selection on the indexes because I believe that a stronger adherence to the 1996 Task Force recommendations improves the strength of the manuscript.

Although STV and LTV are mainly used for fetal heart rate variability analysis, they use original methods to assess short term and overall variability of the signal. ,These methods could have bring supplementary information (than RMSSD and SDNN) and we firstly found interesting to study these indexes in a different clinical context than fetal heart rate monitoring. 

As requested we have clarified why we used LTV and STV. We have modified the paragraph focusing on the LTV and STV as follows (lines 166-168): 

“Although STV and LTV are mainly used for fetal heart rate variability analysis, they use different methods than SDNN and RMSSD to assess the overall and short term variability of the ANS and could therefore provide additional information in the present clinical context.”

5. Lines 357-358: add a reference to ApEn and SampEn.

This reference has been added for ApEn and SampEn: “Shaffer F, Ginsberg JP. An Overview of Heart Rate Variability Metrics and Norms. Front Public Health 2017;5. pmid:29034226”.

6. In the limitations section, an interesting future development for the AAGA prediction could be the inclusion in the model of indexes derived from the blood pressure variability and from the bivariate analysis of the baroreflex, which are also affected by anesthesia (see previously mentioned references). Was invasive blood pressure monitoring performed during anesthesia in this database? If not, in future works it could be interesting to test this approach on a different database that includes this signal.

Indeed we did not monitored invasive blood pressure for this study. We added this limitation in the discussion (lines 422-425): 

“In this study we only considered HRV features whereas taking into account the blood pressure variability would allow obtaining a much more realistic model of the baroreflex response to anaesthetic agents [21,22]. In future work, the addition of invasive or non-invasive continuous blood pressure monitoring could allow testing this approach.”

Reviewer #2: Title: Heart rate variability analysis for the prediction of accidental awareness during general anaesthesia: a feasability study

Journal: PLOS ONE

Manuscript ID: PONE-D-24-08727

General opinion

The authors measured various heart rate variability (HRV) variables during propofol-remifentanil anesthesia in clinical trials and tested the ability of an HRV-based model to predict “BISTM≥60” periods. They want to confirm that HRV analysis can be used to predict the occurrence of accidental awareness during general anesthesia. However, both experimental design and result analysis are far from achieving this hypothesis. The pictures are vague even the download version, some methods are not clear and no deeper analysis at all. Therefore, I do not recommend the publication of this paper. Comments are as follows:

Major comments

1. The title of this paper is overstated, they only used the ' propofol-remifentanil ' anesthesia program, which doesn’t represent all cases of general anesthesia.

We agree that the paper title was overstated and we modified it as follows: 

“Heart rate variability analysis for the prediction of pre-arousal during propofol - remifentanil general anaesthesia: a feasibility study”

2. The introduction section should include more recent studies on the emergence of accidental awareness during general anesthesia.

We added a more recent reference in the introduction as requested: 

“Kim MC, Fricchione GL, Akeju O. Accidental awareness under general anaesthesia: Incidence, risk factors, and psychological management. BJA Educ. 2021 Apr;21(4):154-161. doi: 10.1016/j.bjae.2020.12.001. Epub 2021 Jan 21. PMID: 33777414; PMCID: PMC7984969.”

3. The narrative of the material and method part is not concise enough.

We agree with the reviewer that the material and method part may seem long. However, several of the other reviewers requested clarification making it impossible to reduce this paragraph. In addition, we believe that a good understanding of the paper makes it necessary to provide a complete description of the different stages of the protocol.

4. The writing was confused, and the methods and results do not correspond to each other.

Several modifications in the method and the results were made making these two parts more understandable.

5. The initial clinical trial included only 52 patients, and the sample size was too small to be convincing. 

Indeed, this study was a retrospective analysis of another study dataset. We added this limitation in the discussion paragraph as follows (lines 414-415):

“Particularly, the sample size was small and not computed in accordance to the primary objective of this study. These results need to be confirmed on a larger population in a prospective clinical trial.”

6. In the discussion section, they pointed out that they only analyzed simple, well-known time-domain and spectral HRV variables, the inclusion of more complex, non-linear indexes (e.g. the approximate entropy or the sample entropy) would perhaps have led to better results. It is recommended to add this part of the work in this study.

As specified in the manuscript, we decided not to include non-linear indices in this analysis because they are complicated to interpret from a physiopathological point of view. In addition, they are often estimated over relatively long time windows, making them poorly compatible with our study design.

Minor comments

1. Please add the graded title before each part of the article.

We added some graded title throughout the manuscript

2. The abstract part is not clear about the results and conclusions of this study.

As suggested, we clarified the abstract. 

3. Please unify the style of the table.

The style of the tables has been unified. 

4. Please adjust the order of images and text, don’t place the figure at the end of the article!

Journal's guidelines ask to submit image separately.

5. The resolution of the image is too low.

We have modified the resolution of the image.

6. It would be better to include additional latest references.

As suggested we added more recent references in the manuscript. 

7. Please revise the format of the references, the page number of the reference was listed (Page 23, line 385).

We revised the format of the references.

Reviewer #3: The research procedure is clear, and the result figure of the restricted cubic spline function can be added to the result part. 

We added the cubic spline function in the results part and modified the results paragraph as follows (lines 274-276):

“Shape of associations between each parameter and awakening were shown in Fig 2. There is no major deviation in log-linearity assumptions for all parameters, except for SDNN and LF where log-transformations were needed.” 

The discussion part fails to explain well why HRV is used to evaluate the accidental awareness of general anesthesia. The figure at the end of the manuscript is not clear enough.

As suggested by other reviewers we changed accidental awareness by pre-arousal which makes the discussion more clear. 

Reviewer #4: The article explores the feasibility of using heart rate variability (HRV) analysis to predict accidental awareness during general anaesthesia (GA). This pilot study addresses a significant complication in anaesthesia practice, where accidental awareness remains relatively frequent despite continuous electroencephalographic monitoring. By analyzing RR intervals and the Bispectral IndexTM (BISTM), the researchers aim to identify HRV parameters indicative of patient arousal.

The study demonstrates an innovative approach by leveraging HRV analysis, which is less commonly utilized in this context, to address the issue of accidental awareness. The methodology is comprehensive, employing both multivariate logistic regression and classification and regression tree algorithms to provide a robust analysis framework.

However, some issue that requires attention.

The phrase "Given that EEG signals are influenced by the brain’s cortex only and that awareness and connectedness result from interactions between subcortical structures, it is not surprising that EEG signals do not reflect the activation of the subcortex, which leads to awareness during surgery" requires clarification and proper referencing.

Firstly, the assertion that EEG signals are influenced by the brain’s cortex only is inaccurate. EEG signals, while predominantly reflective of cortical activity, can also capture electrical activity originating from subcortical structures through volume conduction. Furthermore, the claim that EEG signals do not reflect subcortical activation lacks a supporting reference.

As suggested by the reviewer we clarified and added proper referencing to the phrase, as follows (lines 85-95): 

“Simplified EEG signal used for depth of hypnosis monitoring during general anaesthesia rely on a limited number of frontal electrodes, which capture the electrical potential of cortical and subcortical layers [8]. Propofol mediated unconsciousness has been related to an alpha/delta pattern and slow oscillations in the EEG [9,10], but some authors found evidence that connected consciousness sometimes occurs in spite of these patterns being present [11]. Sensory disconnection and loss of consciousness may have different EEG spectral markers, with anterior and posterior cingulate regions probably playing a major role in consciousness [12].”

ECG signal processing

I wonder whether and how the RR intervals were interpolated before applying spectral analysis? If interpolation was used, what kind of interpolation method was employed? Furthermore, how were potential ectopic beats treated during the analysis? Clarification on these methodological details is important for the reproducibility and reliability of the study's findings.

RR interval were resample using a linear interpolation. Before RR series resampling, any ectopic beat or RR series artifact were detected and replace using a specific RR series filtering algorithm developed at the Lille university hospital. 

We modified the ECG signal processing paragraph as follows (lines 138-139 and lines 146-148): 

“Ectopic beats and RR series artefact were detected and replaced by a linear interpolation using a specific algorithm [26,27]. After RR series artefact filtering […] LF was computed via a wavelet transform (e.g. Daubechies 4-wavelet) on a 64-second window resampled at 8 Hz using a linear interpolation.”

Statistical analysis

The phrase "Highly correlated variables were then identified using a principal component analysis (PCA) and then varimax rotation" is not entirely clear and may reflect a misunderstanding of PCA's purpose. To my knowlage, Principal component analysis (PCA) is not primarily a method for identifying highly correlated variables. Instead, PCA is a dimensionality reduction technique that transforms a set of possibly correlated variables into a smaller number of uncorrelated variables called principal components. These principal components are linear combinations of the original variables and aim to capture the maximum variance in the data.

While PCA can reveal underlying patterns and relationships in the data, it does not directly identify which variables are highly correlated. Instead, it identifies combinations of variables (principal components) that account for the most variance. Varimax rotation, a method used after PCA, is applied to the principal components to make the output more interpretable by maximizing the variance of the squared loadings of each factor, thereby simplifying the structure. 

We agree that PCA is not a method directly dedicated to the identification of correlations between variables, since it is a dimension reduction method. In our study, we performed a PCA with Varimax rotation to identify groups of variables correlated with principal components. If several variables are strongly correlated with a principal component, they are often correlated with each other and the true correlations can be quickly checked on the correlation matrix. In our study, a crucial step is to reduce the number of candidate explanatory variables, since there are potentially collinear variables. Our decision to exclude a particular variable was based primarily on Pearson's correlation coefficients. PCA was only used to identify groups of potentially correlated variables.

We modified the statistical paragraph as follows (lines 193-214): “The normality of distribution was assessed using histograms and the Shapiro-Wilk test. Quantitative variables were expressed as the mean ± standard deviation when normally distributed and as median [interquartile range (IQR)] when not. Qualitative variables were expressed as the number (percentage).

In order to identify highly correlated variables, a principal component analysis (PCA) followed by a varimax rotation was applied on all the available variables. Pearson’s correlation coefficients between quantita

---

## [Decision Letter · Decision Letter 1]

5 Sep 2024

Heart rate variability analysis for the prediction of pre-arousal during propofol-remifentanil general anaesthesia: a feasability study

PONE-D-24-08727R1

Dear Dr. Wojtanowski,

We’re pleased to inform you that your manuscript has been judged scientifically suitable for publication and will be formally accepted for publication once it meets all outstanding technical requirements.

Kind regards,

Ernesto Iadanza

Academic Editor

PLOS ONE

Reviewers' comments:

Reviewer's Responses to Questions

**Comments to the Author**

1. If the authors have adequately addressed your comments raised in a previous round of review and you feel that this manuscript is now acceptable for publication, you may indicate that here to bypass the “Comments to the Author” section, enter your conflict of interest statement in the “Confidential to Editor” section, and submit your "Accept" recommendation.

Reviewer #1: All comments have been addressed

Reviewer #3: (No Response)

Reviewer #5: All comments have been addressed

2. Is the manuscript technically sound, and do the data support the conclusions?

Reviewer #1: Yes

Reviewer #3: Yes

Reviewer #5: Yes

3. Has the statistical analysis been performed appropriately and rigorously? 

Reviewer #1: Yes

Reviewer #3: Yes

Reviewer #5: Yes

4. Have the authors made all data underlying the findings in their manuscript fully available?

Reviewer #1: Yes

Reviewer #3: Yes

Reviewer #5: Yes

5. Is the manuscript presented in an intelligible fashion and written in standard English?

Reviewer #1: Yes

Reviewer #3: Yes

Reviewer #5: Yes

6. Review Comments to the Author

Reviewer #1: (No Response)

Reviewer #3: In the manuscript titled Heart rate variability analysis for the prediction of pre-arousal during propofol-remifentanil general anaesthesia: a feasability study contains some interesting findings and are valuable for the understanding of HRV's role in the prediction of pre-arousal.Therefore,MINOR revision has to be done before this manuscript could be accepted for publication in the PLOS ONE.

1.In the discussion section, it is recommended to further discuss the specific changes of HRV under different anesthetic states, especially the physiological significance of LF and HF spectra and their relationship with the depth of anesthesia.

2.The interpretation of various HRV variables can be refined.

3.The clarity of the image and the beautification of the form are also important.

Reviewer #5: Dear Authors,

I would like to express my sincere appreciation for your thorough and thoughtful responses to the concerns I raised. Your revisions have effectively addressed all the issues I identified, and I commend you for the clarity you have brought to the manuscript. I am confident that these improvements have significantly enhanced the quality of your work.

I have only noticed a small error regarding the TCI model referenced as Schiller; I believe it should be changed to Schnider.

Thank you once again.

Best regards.

7. PLOS authors have the option to publish the peer review history of their article (what does this mean?). If published, this will include your full peer review and any attached files.

Reviewer #1: No

Reviewer #3: **Yes: **Duan Xiaoxia

Reviewer #5: **Yes: **Cristian Aragón-Benedí

---

## [Editor Report · Acceptance letter]

10 Sep 2024

PONE-D-24-08727R1 

PLOS ONE

Dear Dr. Wojtanowski, 

I'm pleased to inform you that your manuscript has been deemed suitable for publication in PLOS ONE. Congratulations! Your manuscript is now being handed over to our production team.

Kind regards, 

on behalf of

Dr. Ernesto Iadanza 

Academic Editor

PLOS ONE